# Circular Economy and its Comparison with 14 Other Business Sustainability Movements

**Gergely Tóth**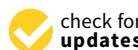

Department for Alternative Economics, Institute of Finance and Accounting, Faculty of Economic Science, Kaposvár University, 7400 Kaposvár, Hungary; toth.gergely@ke.hu

**Abstract:** Circular economy is not the first, and probably not the last "movement" in the arena of sustainability macroeconomic and business solutions. In this article we produce a—not full—list of similar movements from the 1990s, publish a comparative table and propose a simple framework to decide the significant points of the life cycle of such a kind of movement. For significant points and statistics, we use simplified content analysis from normal and scientific research engines. Finally, we use this framework to make a forecast about time for the circular economy approach "to stay on the top" and conclude if these movements are "Much Ado about Nothing" or they help us on our way to a sustainable planetary, social and economic system.

**Keywords:** business sustainability movements; circular economy; life cycle; sustainable development; human economics

---

## 1. Introduction: Hypes, Movements, Scientific Schools

Circular economy and sustainable development goals of the UN are far the most popular topics in the last two years in the business sustainability arena. This was not the case five years ago. It is an interesting question to try to forecast whether it will be the same in 2–5 years' time. To decide that, in this paper we will look at the popularity of other similar movements. We will examine the hypothesis that these movements come and go as fashion, or they keep up the interest for sustainable development, as a whole approach, to prepare a paradigm change from unlimited growth to sustainable development.

The first sentence of the Book of Genesis (and the whole Bible) is: "In the beginning God created the heaven and the earth". If once the "bible" of the modern business-environmental movements would be written, it could start with such a sentence: "In the beginning environmentalist created Recycling!"

Indeed, recycling is a very old approach. Written sources mention paper recycling from 1031 from Japan [1,2], and the second utilization of resources were used as nation-wide strategies in World War 2 UK and USA. However, recycling, as an "environmental movement" or techno-scientific approach is much younger. The first mention of recycling in *Google Scholar* is from the early 1800s, and we have altogether 1730 records till 1900. Currently it is about 20–60 thousand in a single year (32,500 Google Scholar hits between 1 January 2018 and 26 June 2019, 22,800 between 1 January and 26 June 2019, 308,000 since 1 January 2015.) and 2.8 million in total.

If we look at (normal) *Google* hits, recycling is a very popular topic. Its product life cycle is similar to Coca-Cola from marketing (*Life cycle* and *life cycle assessment* is used in two meaning in this article. The main meaning is from marketing science: How long a product can stay in the market, before it gets technically or fashionwise obsolete. Some exceptional): The product does not get obsolete, it does not go out of fashion, it stays on the top. If we hit the research phrase "recycling" to Google, we get approximately 332 million results (This article heavily relies on normal Google searches, Google Science hits and time series of hits etc. from *Google Trends*. It is necessary to deduce the "noise" and

severe short-term time fluctuation of results. For this reason, all hits are from a period of three days: 27–29 June 2019, unless otherwise indicated). With these numbers, recycling is far the most popular movement among the 15 we are considering in this article, its dominance *in everyday use* is "oppressive" and irreversible, *in scientific publications* it is only highly outstanding and unquestionable.

This is the point to explain why we use the term *movement*. We could call these fifteen things *hype*, as they have characteristics of fashion, people are enthusiastic about them, but then they go out of fashion. However, they are too well supported and scientific to be called a hype. We could also call them *scientific schools*, as they are well defined, we have scientific evidence behind them in forms of monographs [1,3], primary research [4,5], journal articles [6–8]. For example, literature supports that circular economy can contribute to the energy [9] and material [10,11] perspectives, embracing topics from residential photovoltaic systems [9] to sewage sludge biogas solutions [12]. New movements are widely documented with systematic literature reviews [13,14].

However, most of these studies come from semi-scientific sources like the consultancy sphere [15] or the European Commission [16]. These institutions—although making excellent and reliable research with hard work—have a primary interest to spread what they consider good politically, and these forecasts are often positively biased. So, these things do not show the characteristics of scientific schools in the long run, they might be called one scientific school *(the business sustainability school)* in the long run. It is also often the case that a thing has a look of a scientific school or looks like a hype, but then another characteristic of it becomes more dominant. Marxism is an example for that: If it did not turn into a social movement with the aim to change the world very pragmatically, we would probably consider Marxism as one of the most elaborate schools of economics. However, the political movement faded this characteristic of being a scientific school.

We could also look for other expressions like *paradigm*, *meme*, *program*, etc., but we find that the connotation of the world of *movement* is the most proper for our purposes. This is the strongest common term but saying that we do not ignore that the 15 movements have different characteristics. For instance, *recycling*—apart from being a movement—is a very practical approach to waste management, *zero emission* is mostly known in the car industry, *cleaner production* is a very highly ranked scientific school with an excellent dedicated journal, and so on. For the sake of simplicity and our intention to compare these things, we call them movements. So, let us see, what similar movements can we consider as predecessors of the circular economy.

## 2. Dataset: A Catalogue of 15 Business Sustainability Movements

We can pick a list of 15 movements showing similar characteristics to circular economy. They often have common fields, so in order to define them we describe them shortly, to have a common understanding. In the list below (Figure 1) we use the most widespread definition and one-paragraph description of the movement, if it is not available from secondary source, we produce a short summary. In some cases we put the most well-known symbol or "founding father" (namely 1. *Recycling* logo, 3. *Cleaner Production*—UN logo, 10. *Corporate Social Responsibility* explaining graphic, 11. Günther Pauli with *Blue economy*, 12. Michael Porter with *Creating shared value*, and a 15. *Circular economy* explaining chart, referring back to the beginning of the list: product life cycle). This list could be extended with phrases like eco-efficiency or eco-design, but a list of 15 significant movements is strong enough to see differences, commonalities, and most of all meet our primary goal: To depict the life cycles.

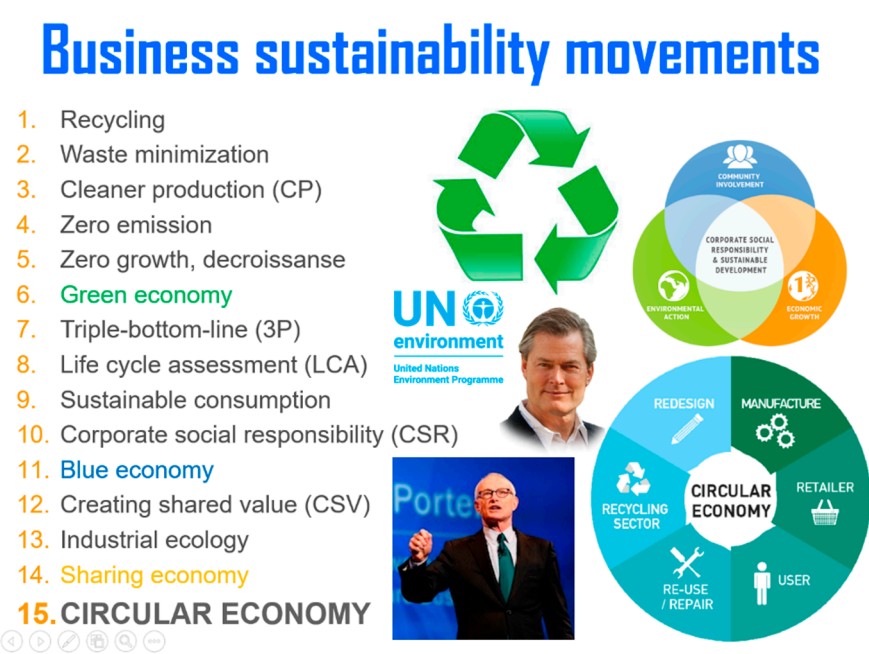

**Figure 1.** The 15 business sustainability movements in our focus.

## 2.1. Recycling

Recycling is a procedure to convert waste materials into useful objects again, that is to produce new products from old (vs. so-called virgin) material. Most common examples are paper, glass, and metal recycling. Compound products are harder to recycle, cars or electronics are made of a number of carefully combined materials, which does not ease detachment and reutilization. Recycling is normally considered as an environmentally friendly solution opposite to waste disposal (dumping), incineration (utilization of the energy content) is half-way. The waste mitigation hierarchy or the three 'Re' are often cited [3,16] that is Reduce-Reuse-Recycle. In this sense the best environmental solution is (i.) not to produce and consume, than (ii.) to use things for the same purpose without an energy-intensive de- and remanufacturing (e.g., selling mineral water again in the same glass bottles), and (iii.) finally convert material through handicraft or industrial processes into new products. As we will see, recycling is the first and far the best known movement in our list.

## 2.2. Waste Minimization (WM)

Waste minimization is a systematic approach to reduce, and if possible, to prevent the "production" of unintended by-products and other waste material, including fluent and gaseous emissions. Ojovan and Lee [17] defines waste minimization as a process of reducing the amount and activity of waste materials to a level as low as reasonably achievable. WM strongly relies on the waste mitigation hierarchy: reduce-reuse-recycle (as shown in Figure 2). Sometimes other 'Re's are added like *rethink*, *redesign*, *refuse*, *replace*, *reengineer*—but the point is the same, this is mostly playing with the words. As Rosenfeld [18] states, the objective of WM is to decrease the amount of hazardous waste bound for energy recovery, treatment, and disposal facilities. Utilization for the same purpose in the same form (reuse), in a modified form (remanufacture) and in a new form (recycle) is sought instead. Although waste minimization is already mentioned in 1974 [19], it became a massive movement from the 1990s, propagated by prestigious organizations like the US EPA, specialized UN agencies, etc. 1984 the US Congress passed amendments to the Resource Conservation and Recovery Act (RCRA) declaring waste minimization to be national policy [20].

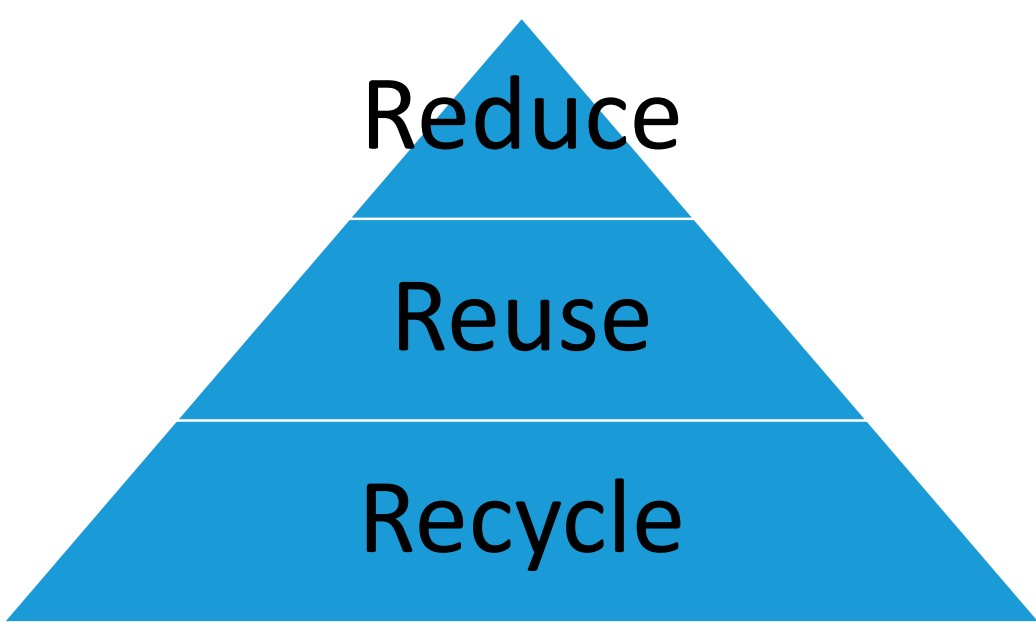

**Figure 2.** The 3 'Re's—the waste avoidance/utilization hierarchy.

*2.3. Cleaner Production (CP)*

The methodology, earlier also termed pollution prevention, is based on preventive solutions as opposed to end-of-pipe technologies. Besides being logical it has also been proved by several studies that if a procedure is originally formulated so as not to create pollution or waste it is not only environmentally positive, but also financially advantageous. This way materials and energy obtained at high costs are not wasted by low efficiency. In contrast, end-of-pipe solutions leave production processes unchanged, but add supplementary devices, e.g., filters, cleaners, to them. These supplements have extra cost on the one hand, and on the other many times just transform one type of pollution into another (e.g., Sludge, energy-plant ash). They are of course needed and handy in everyday practice, but our main perspective should be prevention. Cleaner production is propagated through the international network of CPCs, Cleaner Production Centers.

The promotion of energy efficiency can be taken as a special manifestation of cleaner production. Here our aim is to keep wasted energy at the lowest possible level at an organization or in a building. As a result of CIPEC (Canadian Industry Program for Energy Conservation), for example, 5000 companies, responsible for 98% of the total industrial energy consumption, decreased their energy intensity by 9.1% between 1990 and 2004. Energy conservation is usually attained by the combination of two types of measures: "hard" measures are technological changes (like recuperative devices, installation and reuse of thermal energy waste), while the "soft" ones request behavioral or administrative modifications only. Experience shows that at least half of the environmental problems would be prevented by responsible behavior. Looking at it from another angle, the development of technologies will never be an answer to mankind's environmental problems by itself, to reach this goal we have to change our own behavior [21].

*2.4. Zero Emission*

Zero emission is a well-researched topic and its connection to other movements like CP or LCA are apparent in the literature [8,22]. Some even assume that this approach could be a holistic tool to bring about a sustainable society [23]. Nevertheless, the most well-spread use of the term is in the automotive industry, hinting that zero emission is a narrow focused methodology referring to an industrial or mechanical process, motor, or engine, emitting no waste products of any kind that pollute the environment or contribute to climate change. Nieminen [5] shows that this approach is very closely linked to best available technologies (BATs), eco-efficiency and LCA. A complex approach

to zero emissions was first published in 2002 [24] (Dixon, Porche and Kulick), but much earlies it gave birth to ZERI—Zero Emissions Research and Initiatives in 1994. The movement than was reborn in the Blue Economy movement by the same think-tank, Günter Pauli.

## 2.5. Zero Growth, Decroissanse

Actors of the business sphere are more practical minded than to be easily put off by some conceptual obscurity about how to define sustainability in every-day use. Especially because from the 60's they have been susceptible to strong attacks first in the name of environmental protection, then sustainable development. Some even started talking about zero growth as the practical realization of sustainable development [25,26]. Zero growth is obviously contrary to the growth myth running in the blood of both micro and macro level decision makers in economy [21,27].

## 2.6. Green Economy (GE)

The green economy can be defined "as economy that aims at reducing environmental risks and ecological scarcities, and that aims for sustainable development without degrading the environment" [28,29]. GE is closely related with environmental and ecological economics, but it has a more politically applied focus. Although the UN Environmental Program adapted the idea, its concept is at least more than two decades older: David Pearce, a prominent environmental economist published his report entitled "Blueprint for a Green Economy" already in 1989 [30]. The book had been prepared by the London Environmental Economics Centre (LEEC), a joint venture by the International Institute for Environment and Development (IIED) and the Department of Economics of University College London (UCL). The Pearce Report demonstrated models where environmental elements in threat of being polluted can be costed. The green economy concept urges systems of taxation which would both reduce pollution by making it too costly and generate revenue for cleaning up the damage. A central GE concept is therefore "the polluter pays" principle.

## 2.7. Triple-Bottom-Line, Alias 3P

Big enterprises made up their own well operationalized concept of sustainable development ("Triple bottom line" also used as TBL, 3BL, People, Planet, Profit, originates from John Elkington, the influential English founder of SustainAbility, from 1994 [31]). As a matter of fact—though not to the satisfaction of all—consensus is about to be reached on the basis of "something is better than nothing". According to this corporate sustainability is the outcome of a triple optimization, or "triple bottom line". It is a three-legged model in which the foundations are the three columns of ecological, social and economic sustainability. The operationalization of corporate sustainability usually means that eco-efficiency is taken for ecological responsibility, keeping to basic norms (such as improving working conditions, giving financial aid, not using child labor and abuse) stands for social sustainability and economic sustainability is clearly understood as the enterprise's long term profitability [21,31].

## 2.8. Life Cycle Assessment (LCA)

The method of Life Cycle Assessment embraces environmental impacts of the product during all stages of its life-cycle. Such an assessment contains all the in- and outgoing material and energy flows separately in the phases of the production of raw materials, processing/manufacturing, usage and becoming waste, not forgetting to consider the transportation linking these phases. Once we have drawn the "boxes" representing these processes (which might amount to thousands within a somewhat more complicated industrial framework like that of manufacturing automobiles) and their input-output flows, we can proceed to summarize the impacts using natural indicators, ending up with an eco-balance. Here we can apply different methods to adapt the different measures into comparable measurement units. Available software (e.g., Gabi) can be of great help, especially because of their evaluation methods in the background (e.g., BUWAL). The major steps of LCA are setting the system limits, inventory analysis and, finally, impact assessment. A number of ISO 14,000 standards deal with LCA [21].

### 2.9. Sustainable Consumption

Sustainable consumption and production aim to promote resource and energy efficiency, sustainable infrastructure. Its strategic goal is to provide access to basic services, green and decent jobs and a better quality of life for all. It is one of the 19 Sustainable Development Goals (SDGs) of UN by 2030, under the name "responsible production and consumption" [32]. Already in 1992, at the United Nations Conference on Environment and Development (UNCED) the concept of sustainable consumption was established in chapter 4 of the Agenda 21. In 2002 a ten-year program on sustainable consumption and production was created at the World Summit on Sustainable Development in Johannesburg. The definition proposed by the 1994 Oslo Symposium on Sustainable Consumption [33] defines it as "the use of services and related products which respond to basic needs and bring a better quality of life while minimizing the use of natural resources and toxic materials as well as emissions of waste and pollutants over the life cycle of the service or product so as not to jeopardize the needs of future generations" [33].

### 2.10. Corporate Social Responsibility (CSR)

It is written in the EU Green Paper on CSR [34] "most definitions of corporate social responsibility describe it as a concept whereby companies integrate social and environmental concerns in their business operations and in their interaction with their stakeholders on a voluntary basis." The Commission recognizes that CSR "can play a key role in contributing to sustainable development while enhancing Europe's innovative potential and competitiveness" [34–36]. According to EU initiatives enterprises "over comply" legislation in collaboration with their stakeholders.

According to the WBCSD "Corporate social responsibility is the continuing commitment by business to behave ethically and contribute to economic development while improving the quality of life of the workforce and their families as well as of the local community and society at large" [37] (p. 6).

The so-called "deep" definition for CSR is the following: "The Truly Responsible Enterprise (i) sees itself as a part of the system, not a stowaway concerned only about maximizing its own profit, (ii) recognizes unsustainability (the destruction of natural environment and the increase of social injustice) as the greatest challenge of our age, (iii) accepts that according to the weight they carry in economy, governments and enterprises have to work on solutions, (iv) honestly evaluates its own weight and part in causing the problems (it is best to concentrate on 2–3 main problems), and (v) takes essential steps—systematically, progressively and focused—towards a more sustainable world" [21].

### 2.11. Blue Economy

The Blue Economy concept was officially laid down in the same titled book of Günther Pauli in 2010 [38], but it refers back to the Zero Emission movement by the same author. It began as a project to find 100 of the best nature-inspired technologies that could affect the economies of the world, with the condition of providing basic human needs—potable water, food, jobs, and habitable shelter—in a strictly sustainable way. Hundreds of technical innovations were found and described, that could be bundled into systems functioning similar to ecosystems.

### 2.12. Creating Shared Value (CSV)

Creating shared value is the latest "hype" in our catalogue, it was first introduced in an often-cited Harvard Business Review article *The Link between Competitive Advantage and Corporate Social Responsibility* [39]. The business concept was proposed by Michael E. Porter, a leading authority on competitive strategy and head of the Institute for Strategy and Competitiveness at Harvard Business School, and Mark R. Kramer, Kennedy School at Harvard University and co-founder of FSG, "a mission-driven consulting firm". The main premise behind CSV is that of "extended CSR". Authors are very ambitious about their concept: They promise CSV has the power to unleash the next wave of global growth and to redefine capitalism [39]. On the other hand, critics say that "Porter and Kramer

basically tell the old story of economic rationality as the one and only tool of smart management, with faith in innovation and growth, and they celebrate a capitalism that now needs to adjust a little bit". They regard CSV as a "one-trick pony approach" with very little chance that an increasingly critical civil society would buy into such a story [40]. It is not clearly explained, if the current income from products in the market are not shared in a moral, just way, why would this happen in the case of "creating more value" (basically increasing retail prices due to more value added). This is not Porter's first approach, he basically connects competitiveness with many trendy approaches, like efficiency, the environmental cause or CSR [41].

### 2.13. Industrial Ecology

Industrial ecology aspires further than cleaner production since its ambition is not the optimization of a specific process, but the creation of a certain industrial eco-system. Here the waste produced by a process or a factory is the base material for another. Its tool kit does not contain too many new elements though, but, besides recycling, is made up of the same as that of the forenamed cleaner production, life cycle assessment and eco-design [21].

### 2.14. Sharing Economy

In the sharing economy, persons rent or "share" things like their cars, rooms, houses or apartments to other people. Also, personal time is not sold, but shared in a peer-to-peer fashion [42,43]. Sharing economy is a basically new approach to the ownership, use and marketing of products and services and has the highest chance to turn the current form of market economy into something slightly or dramatically different. The term is used to describe distributing goods and services differently from the traditional business models via hiring employees and selling products to consumers (as depicted in Figure 3). Others call it "access-economy", which might be a more proper, but less used term [7]. Uber and Airbnb are just two iconic examples of the sharing economy, generating massive and fierce debate among professionals, regulators, and researchers. Sharing economy is not fully dependent, but in its current form heavily relies on Internet-based social networks: a feeling of trusting—formerly unknown people through—the Network substitutes the traditional feeling of trusting your friend, group member or other peer in the local, personal society.

**Figure 3.** The bright future of sharing economy by PWC—from 5% market share in 2013 to 50% in 2025.Source: PricewaterhouseCoopers, 2014 [15].

*2.15. Circular Economy*

According to the definition of Merli, Peziosi and Acampora [13] circular economy "aims to overcome the take-make-dispose linear pattern of production and consumption, proposing a circular system in which the value of products, materials and resources is maintained in the economy as long as possible". Kirchherr, Reike and Hekkert [14] analyses 114 CE definitions and conclude that it is most frequently associated with a combination of reduce, reuse and recycle activities, which they held a mistake, lacking a systemic shift towards social equity and sustainable development. I agree with their conclusion that CE must aim at far beyond mainstream goals of economic prosperity, at a paradigm shift towards sustainable and human development. The concept of CE can be traced back to the works of David Pearce 1989 [30], Kenneth Boulding 1965 [44], and Tim Jackson 1993 [45].

After discussing the fifteen movements in detail, let us turn our attention to their common life cycle.

### 3. Method: A Proposed Life Cycle

According to our hypothesis, a hypothetical life cycle of the business sustainability movements can be constructed. They are known and practiced long in history, for example [1,2] mention *paper recycling* from 1031 Japan, *waste minimization* was probably a practice—although not under this name —in all historic times, due resource scarcity and common sense. William Foster Lloyd in 1833 [46], and Garrett Hardin [47], popularizing him in 1968 described the sharing economy in the *Tragedy of Commons*, which was rather the mainstream and not the exception before the massive enclosure in the 18th century England. However, waste minimization and sharing economy did not appear as a comprehensive and broad movement until the recent decades. So "historic times" on Figure 4 can take centuries or millennia, but as a movement, hype, widely spreading business initiative or public policy instrument by the UN, EC and other respected international agencies is normally taking place from the 1990s, when global environmental problems have been commonly understood and accepted. The historic (latent) period and the fashion (explicit) period is depicted on Figure 4 with red turning to green respectively. The figure proposes a life cycle as well: Steady and slowly accelerating growth, peak and going out of fashion, where the horizontal axis is a logarithmic scale.

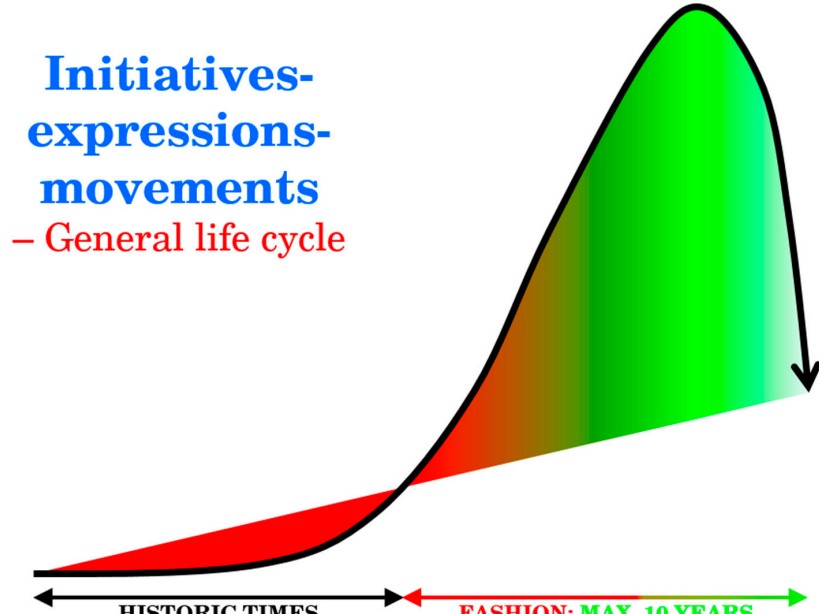

**Figure 4.** A hypothetical life cycle of a business sustainability movement, time (horizontal) and popularity (vertical).

Compering historic practices and modern renaissance of these approaches we could conclude that modern societies keep on reinventing the wheel. What is worse, from the catalogue of the previous section we could conclude that we have reinvented at least 15 different wheels. As we emphasized, these movements we consider one wheel, although varying in shape, material and other important characteristics. Only the business sustainability movement is a wheel, with slight variations.

However, the main purpose of this article is not to create a catalogue of business sustainability movements, but to look at their respective life cycle. Is it true that they really emerge, fly high and disappear? Do they add new peaks and keep up public interest for business sustainability? In the next session we will see, that this hypothesis is only partly true, at least with our methodology: It is easy to be present on the Internet, it is hard to top the hit lists, but what is really impossible to disappear from there.

On Figure 5 we tried to depict a somewhat pessimistic hypothesis: business sustainability movements come, flourish and go. The thin color curves represent recycling, waste minimization, cleaner production, blue economy etc., the heavy grey curve represents the business sustainability movement in general. Colors extend a bit the total life cycle, but unless new hypes come, public interest will turn to other topics, in this accelerated and pulsing era of big data and mass information.

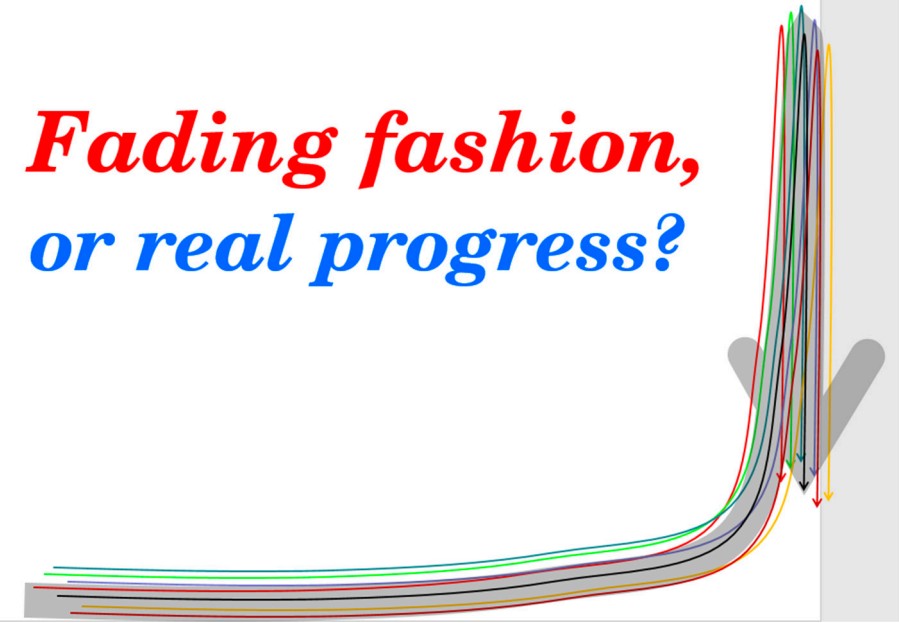

**Figure 5.** The individual LCs and consolidated hypothetical life cycle of business sustainability movements, time (horizontal) and popularity (vertical).

This would mean that we have to produce hypes in every 5–10 years, and repeat the tedious efforts of defining, finding positive examples, publishing handbooks, case studies, technical guides and policy documents, etc. This would also mean that these approaches hardly come to the boardrooms and university textbooks or they disappear very quickly. In the next section we use statistics of Google searches and hits in both the public pages and the scientific arena, in the last fifteen years. Google trends gives and excellent tool to produce time series in all different combinations.

Basically, our method is relatively simple: as we look at fashion and popularity AND presence in scientific publications in parallel, we look at 1) overall Google hits [this is what we call "normal", without any screening] AND 2) hits in qualified scientific databases. The latter is twofold: Google Scholar and Science Direct. Normal and scientific hits normally correlate, but not necessarily: sometimes they show fairly different results (as seen on Figures 6 and 7, e.g., sharing economy is very popular in normal Internet, but not visible in Google Scholar).

## 4. Analysis: The Comparative Table and Citations

In the table below (Table 1) we summarized the main characteristic points of fifteen sustainability movements of business and economics. In column II is the oldest paper in Google Scholar. Columns III–V are calculated values from Google Trends (as of 28 June 2019), showing the respective hits in Google and Google Scholar. We show highest and lowest values and their time (year and month). Columns III–V consider a 15-year period between January 2004 and June 2019. Column VI is again a somewhat anecdotal piece of information, but it is mostly agreed upon and easy to check. In column VII we cross-check Google trends and choose the scientific database over the common one: We decide the approximate length of the movements' fashion based on hits in Science Direct (as of 20 July 2019. We consider a movement "on top", if Science Direct lists minimum 100-300-1000 papers per annum, in relation to the total hits, to keep a balance and add a positive discrimination to less visible movements).

**Table 1.** The comparative table of 15 business sustainability movements—simplified content analysis.

| | I. Movement | II. First sci. mention | III. Top year, G. &G.schol. hits | IV. Bottom year, G.& G.s. hits | V. Google & G. sc. hits 2019 | VI. High as movement | VII. Sci Dir hits **years** on top |
|---|---|---|---|---|---|---|---|
| 1. | **Recycling** | Platon, 4th century B.C. | 04-2008: 100 02-2004: 100 | 02-2015: 58 07-2017: 18 | 06-2019: 85 06-2019: 26 | Since WW2: 1939–1945 | 429,476 **80** |
| 2. | **Waste minimization** | P.R. Taylor, 1974 [1] [19] | 01-2004: 1 02-2004: 6 | 01-2004: 1 01-2004: 0 | 06-2019: 1 06-2019: 1 | 1970s 1980s | 4541 **10** |
| 3. | **Cleaner Production (CP)** | UNEP-UNIDO, 1992 [48] | 02-2004: 1 04-2004: 12 | 01-2004: 1 03-2004: 0 | 06-2019: 1 06-2019: 7 | NCPCs & NCPPs 1994 | 25,567 **22** |
| 4. | **Zero emission** | US Congress, 1970 [49] | 09-2009: 1 01-2004: 13 | 01-2004: 1 04-2004: 0 | 06-2019: 1 06-2019: 1 | ZERI 2004 | 11,849 **12** |
| 5. | **Zero growth, decroissanse** | Meadows, 1972 [50] | 04-2004: 1 04-2005: 7 | 01-2004: 1 02-2004: 0 | 06-2019: 1 06-2019: 1 | OECD 1985 | 3565 **8** |
| 6. | **Green economy** | Pearce, 1989 [30] | 05-2008: 1 04-2004: 11 | 03-2004: 0 01-2004: 0 | 06-2019: 1 06-2019: 2 | ICC 2012 | 2737 **8** |
| 7. | **Triple-bottom-line, alias 3P [2]** | Elkington, 1994 [31] | 02-2004: 1 09-2004: 9 | 01-2004: 1 01-2004: 0 | 06-2019: 1 06-2019: 1 | 2000s: Co. sust. Reports | 3515 **9** |
| 8. | **Life Cycle Assessment** | Vigon, 1994 [51] | 01-2004: 1 02-2007: 7 | 02-2004: 0 01-2004: 0 | 06-2019: 1 06-2019: 1 | US-EPA 2010 | 23,420 **11** |
| 9. | **Sustainable consumption** | Oslo Symposium, 1994 [33] | 01-2004: 1 01-2004: 12 | 01-2004: 13 05-2004: 0 | 06-2019: 1 06-2019: 1 | UN 2000s | 3620 **7** |
| 10. | **Corporate Soc. Responsibility** | Goodpaster-Matth., 1982 [52] | 04-2004: 5 03-2004: 7 | 07-2006: 2 02-2004: 0 | 06-2019: 2 06-2019: 1 | 2000s: Co. CSR reports | 9842 **8** |
| 11. | **Blue economy** | G. Pauli, 2010 [38] | 11-2018: 1 06-2006: 2 | 01-2004: 1 01-2004: 0 | 06-2019: 1 06-2019: 1 | WWF 2018 | 321 **3** |
| 12. | **Creating shared value (CSV)** | M. Porter, 2011 [39] | 06-2004: 1 11-2007: 1 | 01-2004: 0 01-2004: 0 | 06-2019: 1 06-2019: 0 | EC 2010s | 873 **5** |
| 13. | **Industrial ecology** | Frosch-Gallo-poulos, 1989 [53] | 05-2004: 1 05-2005: 23 | 01-2004: 1 01-2013: 1 | 06-2019: 1 06-2019: 1 | 2000s | 5497 **7** |
| 14. | **Sharing economy** | Benkler, 2002 [54], Lessig, 2008 [55] | 10-2014: 1 07-2007: 1 | 01-2004: 0 01-2004: 0 | 06-2019: 1 06-2019: 1 | Last 5 years | 1552 **4** |
| 15. | **CIRCULAR ECONOMY** | Boulding, 1965 [44] Pearce, 1989 [30] Jackson, 1993 [45] | 02-2019: 2 02-2019: 23 | 03-2004: 0 01-2004: 0 | 06-2019: 1 06-2019: 23 | Last 3 years | 5918 **4** |

[1] This date is mistyped in Google Scholar, as 1874. In reality it refers to the foundation of the Kroll Institute for Extractive Metallurgy (KIEM) at the Colorado School of Mines. KIEM focus areas included minerals processing, extractive metallurgy, *recycling* and *waste minimization*. [2] People-Planet-Profit (or **Profit** ... people ... planet?).

Composing the comparative table of the business sustainability movements in a precise way is harder than expected. In column II, should we specify the first historic example? The first proven use of the expression? The first scientific book or article solely devoted to the topic? We used a mixed

approach. For example, even Wikipedia denotes that Platon spoke about recycling 2500 years ago. However, most of much of Platon's and Aristotle's work was lost, the latter for example only survived in Arabic translations and were later translated back to Greek and Latin. Most of the movements, as we keep on emphasizing, refers back to some ancient and modern wise philosophers, scientists. A good example is the last line in the table, where Kenneth Boulding [44], David Pearce [30] and Tim Jackson [45] are referred to as "founding fathers", but also the Tragedy of the Commons (and herewith Hardin 1968 [46] and Lloyds 1833 [47]) are specified as theoretical basics. Anyway, roots and exact "who said first" is not so important, we could refer this question to monographs dealing with the specific movements (e.g., in CSR [21]).

What is more important from our special perspective, is the recent "web-footprint" and scientific records of the movements in question. The first we approximated with the (normal) Google hits of the last 15 years, the second with the hits in Google Scholar and we made a cross check through Science Direct. We specified some characteristics of these time series in the comparative table.

Our analysis also has some deficiencies: for example, in cell 5/III–IV it is hard to believe that *Zero growth* is on the peak and in its lowest mention in a period of three months. The French term *decroissanse* has a more profound, every day meaning—in English *degrowth* is devoted to the movement, the French *decroissanse* also means *decay, decreasing, reduction*. This means we cannot look for Google searches for *decroissanse* without being extremely biased with our results.

Google Trends is an excellent tool for time series analysis (from intervals of days and hours to a maximum period of 15 years), it gives area specific and detailed geographical information. Its main disadvantage that it is primarily for marketing, not for scientific purposes, its main advantage is that it normalizes hits on a scale of 100. This is the scale we used in the comparative table in columns III–V.

The first result is very apparent from the table: Recycling is far the oldest and most searched referred term of all 15 movements. If we put it to the comparative analysis, other movements become almost invisible (although in scientific articles the difference is much smaller). For this reason, we put the five less known and newer approaches on a joint graph (Figure 6). It is obvious, that single prophets (like Günter Pauli behind the *blue economy* or Michael Porter behind CSV) can have a huge added value in marketing, but this is still a short-term and relatively small push. If it is a long-term strategy and a giant agency as UNEP and UNIDO behind *cleaner production*, the effect is harder and longer. Nevertheless, general, easy-to-understand and appealing approaches like *sharing economy* and *circular economy* are the most successful in the evolution of business sustainability movements. Even the whole business sphere with all pioneering multinationals and their sustainability reports can have a relatively small leverage effect compered to this general appeal to the public. In the case of *circular economy*, Google Trends show us another interesting aspect: at one point around 2004–2005, 2 of the 5 related search terms included *Ellen MacArthur*, a champion yachtswoman from England. After retirement from professional sailing (at the age of 34) she established the *Ellen MacArthur Foundation*, a charity that works with business and education to accelerate the transition to a circular economy. One famous individual can do a lot to popularize the public good.

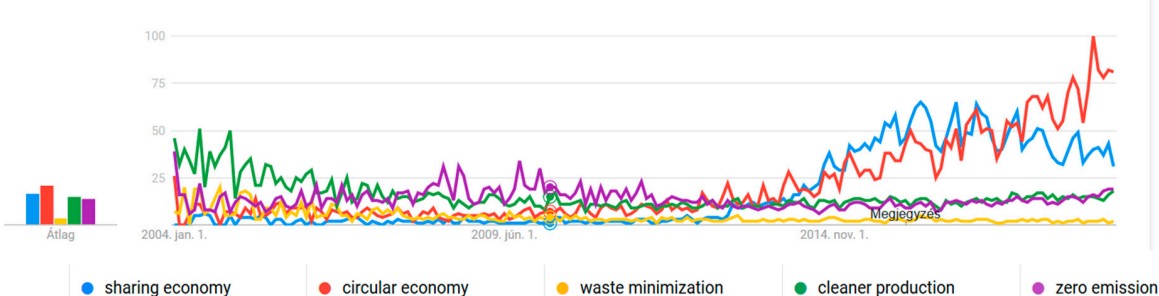

**Figure 6.** Frequency of normal Google searches for the terms *circular economy*, *sharing economy*, *waste minimization*, *cleaner production* and *zero emission.* Data and graph from Google Trends, as of 27 June 2019 (Hits normalized on a scale of 100).

On Figure 6 we see a limited effect of fashion: *Sharing economy* was very popular around 2015, but interest is significantly lost in the last 2–3 years. *Circular economy* was almost invisible till 2013, since that time its carrier is boosting. In science, however, the picture is slightly different. Sharing economy is very little discussed, and cleaner production keeps its positions much better.

We can observe significant regional differences in different countries. As apparent from Figure 8, sharing economy is still almost more popular in Germany, than circular economy. In the USA, the latter has clearly taken over. As well, in a new market economy, like Russia, where sustainability is probably less on the top of the agenda than in the EU or US, we see basically no evaluable activity. *Zero emission* is a more well-known term in the USA than in other countries, probably due to the fact that car development is more regulated by the market in the US, and more by the EC in the European Union (emission standards for passenger cars).

One major conclusion we can already draw here is that instead of competing movements, we should concentrate on strengths of each: *Cleaner production* has a very high scientific literature and technical background through the *best available technics* (BATs), *circular economy* is the newest concept with the contemporarily strongest appeal, *sharing economy* has the highest community (social network and 'apps') support, also there is the most fight around it taking the form of market regulation (Uber vs. taxi companies, Airbnb vs. hotel chains, pirate music sharing vs. traditional recording companies and Amazon, etc.). These fights create significant losses and some bankruptcies but are beneficial for the somewhat halted evolution of modern business towards a sustainable economy.

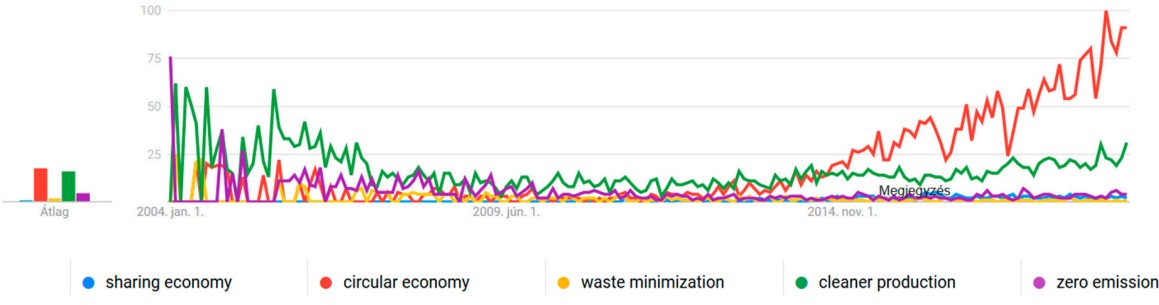

**Figure 7.** Frequency of Google Science searches for the terms *circular economy, sharing economy, waste minimization, cleaner production* and *zero emission.* Data and graph from Google Trends, as of 27 June 2019 (Hits normalized on a scale of 100).

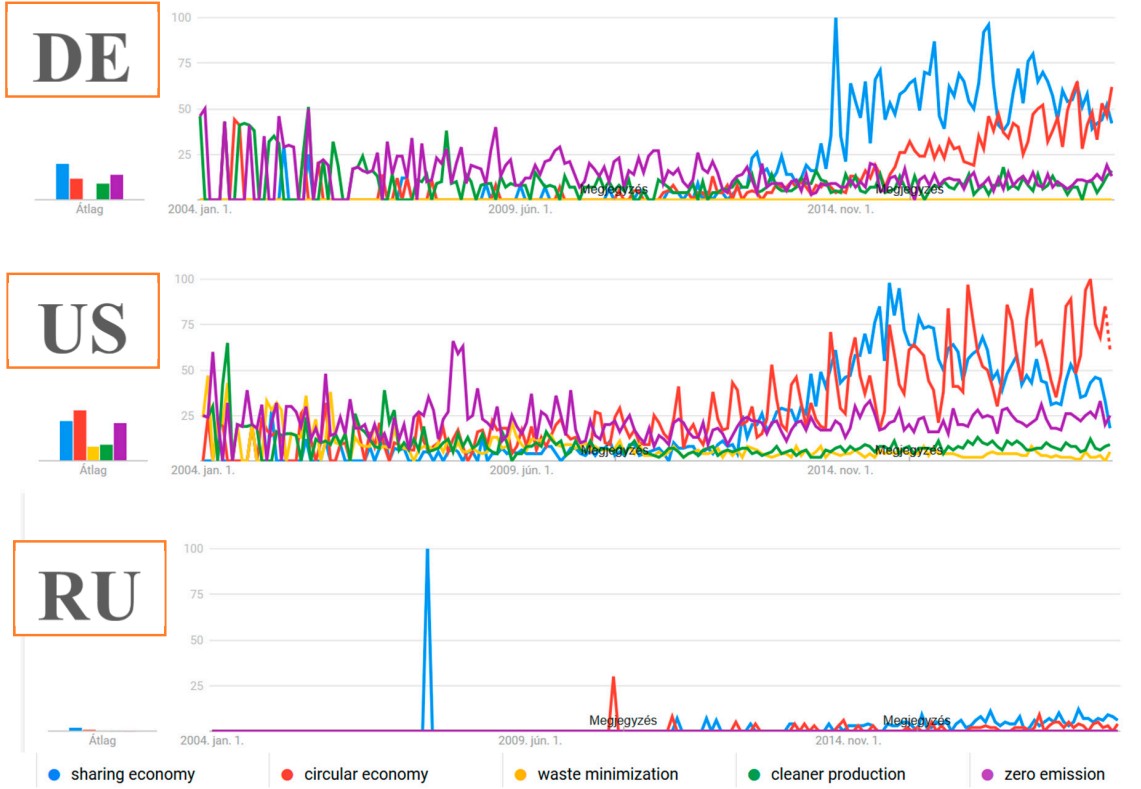

**Figure 8.** Frequency of normal Google searches for the terms *circular economy, sharing economy, waste minimization, cleaner production* and *zero emission* in Germany, the USA and Russia. Data and graph from Google Trends, as of 27 June 2019 (Hits normalized on a scale of 100).

On Figure 9 we can see that *circular economy* is the strongest in Scandinavian countries, South America, South Africa, sharing economy is strongest in Russia, US, core of the EU, Australia. However, a new finding is that cleaner production has very strong support and leads the poll in Brazil and Iran. Instead of looking at these selected pictures, I strongly recommend putting these five phrases to Google Trends, select the 15-year period, and look at individual, interactive maps and charts. If we look at the five individual world maps, one major learning is that the US is strongest in everything, which is connected to the Internet.

On Figure 10 we disclose one of these individual world maps, namely for *circular economy*. Apart from the spatial distribution we also see the most common connected terms, which (including the other 14 movements) could be the topic of further investigation.

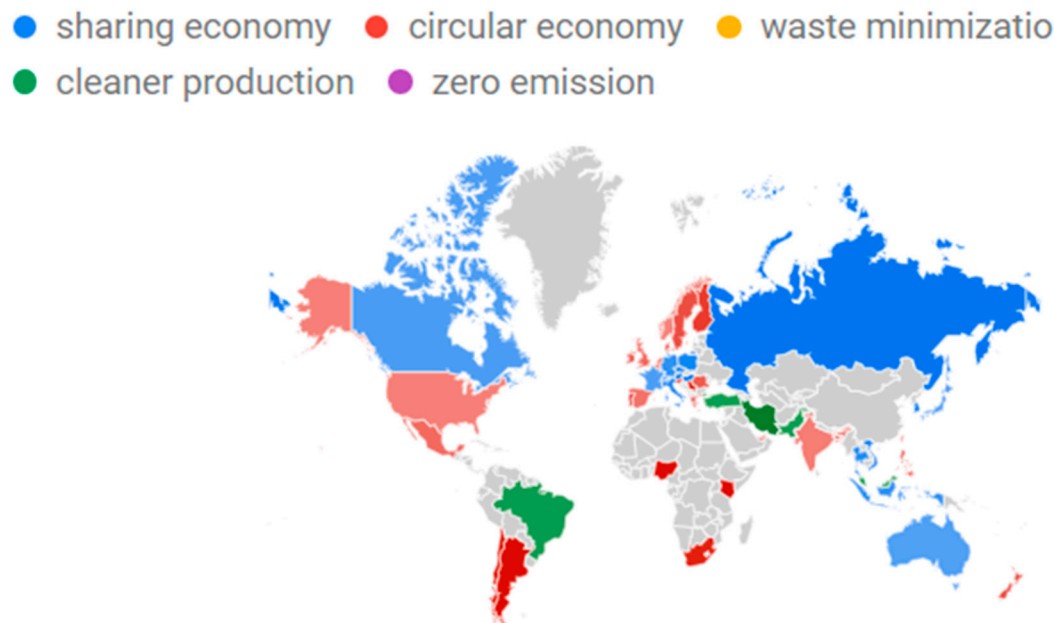

**Figure 9.** Geographical representation of normal Google searches for the terms *circular economy, sharing economy, waste minimization, cleaner production* and *zero emission*. Data and graph from Google Trends, as of 27 June 2019.

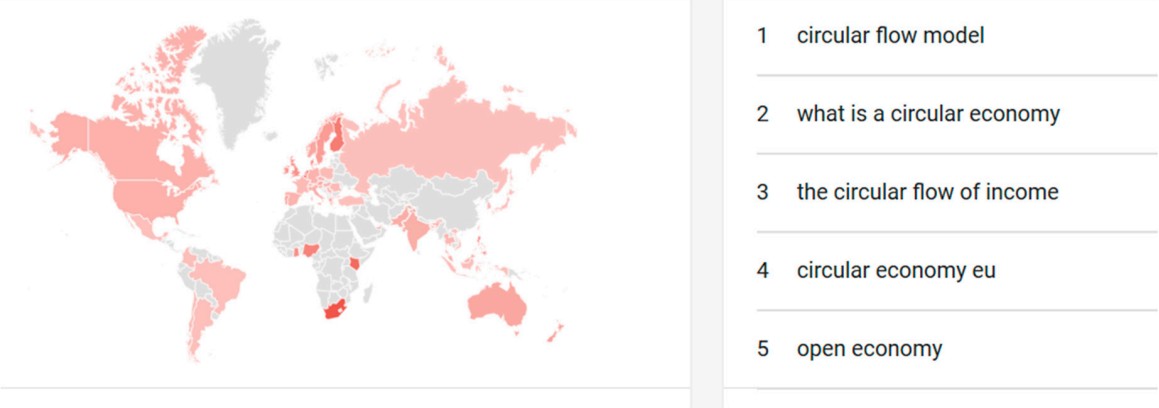

**Figure 10.** Google searches (hits) for the term *circular economy* by regions, and most frequent connected terms. Data and graph from Google Trends, as of 27 June 2019.

In Figures 11 and 12 we compared hits for another set of five of our selected business sustainability movements. As already pointed out, recycling is far most the winner, although its lead is less obvious in Google Scholar than in normal WWW content. In the normal arena, even the second sustainable development is hardly visible (see averages on the left), in science in rare cases it takes over recycling. Recycling is with no question the most technical and least scientific general approach of all.

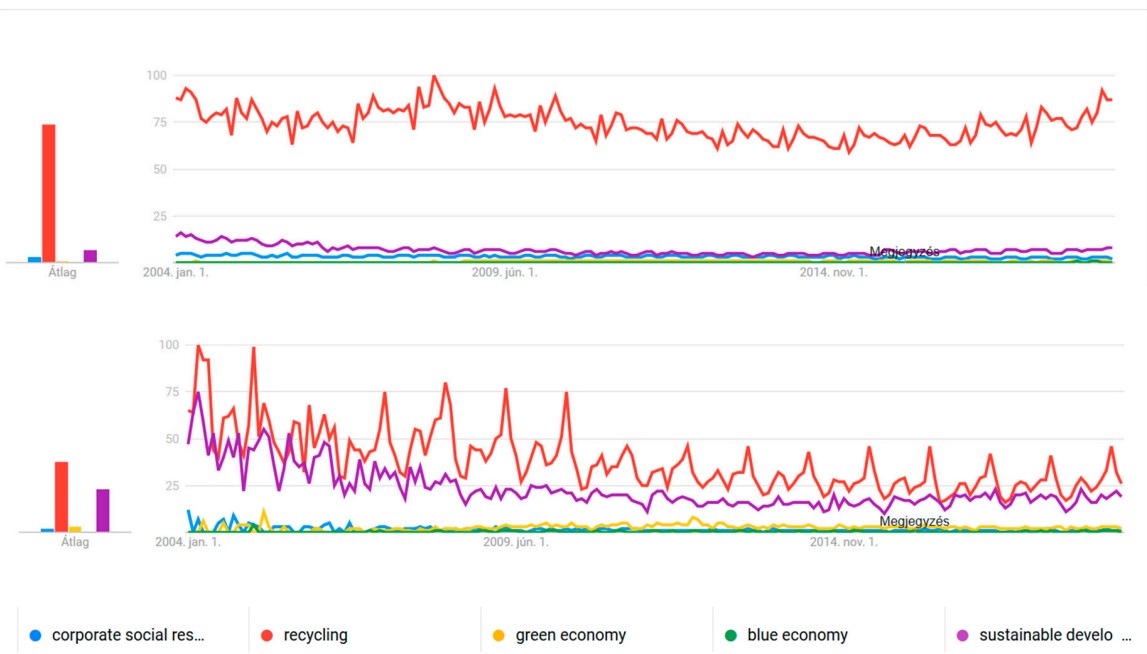

**Figure 11.** Frequency of normal and scientific Google searches for the terms *recycling, sustainable development, corporate social responsibility, green economy,* and *blue economy*. Data and graph from Google Trends, as of 27 June 2019.

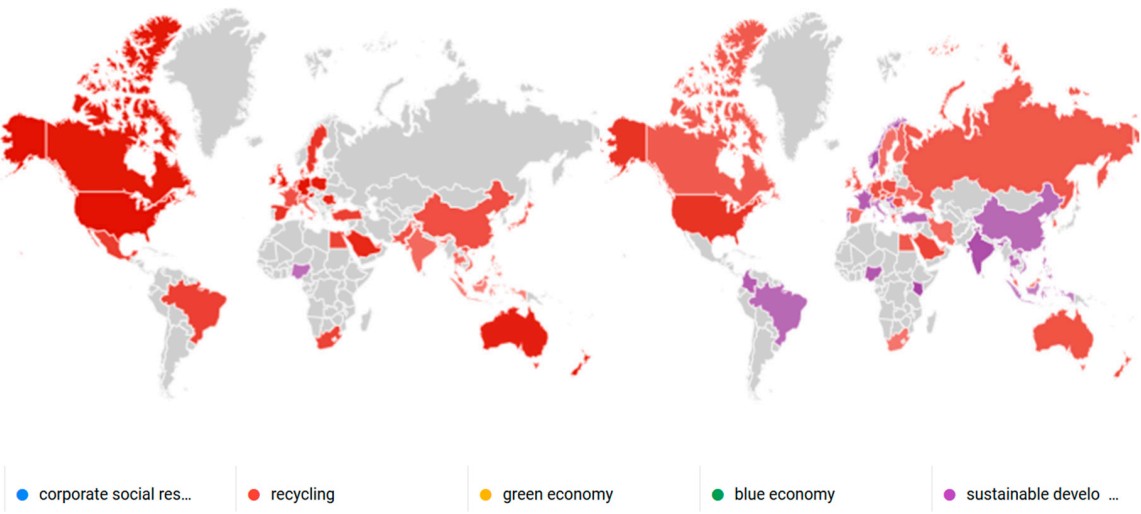

**Figure 12.** Geographical representation of normal and scientific Google searches for the terms *recycling, sustainable development, corporate social responsibility, green economy,* and *blue economy*. Data and graph from Google Trends, as of 27 June 2019.

We could create other graphs and maps, but Google trends has two severe numeric restrictions: It cannot compare more than 5 search phrases on the one hand, and cannot produce logarithmic axis on the other hand, to screen out the powerful dominance of *recycling*. However, we showed a comparative table and massive statistics to justify, modify or falsify our original hypothesis.

At last we can produce a top list of business sustainability movements and draw conclusions (the top 5-6 movements are highlighted in the first column of Table 1). It is remarkable, that we have to use exactly the 5 right search phrases from the 15 potential, and it is also important in what order we type them in to the statistical analyzer. It would be obvious to put the five top terms, but then *recycling* (whose gold medal is not questioned) would fade the other four. So, we look at the comparative table and look for ranks number 2–6, based on the last column: most recent Science Direct hits. We got

a slightly different list from Figures 6–9 (*sharing economy* and *waste minimization* are omitted, LCA and CSR are added). Although circular economy is ranked only sixth in the list of total Science Direct hits, if we consider time—apart from recycling—it leads the list. *Cleaner production* takes the second place now, but it was leading at the beginning of the period (after 2004). It is clear from Figure 13, that they changed place.

Finally, we have to put our vote whether we consider general Google searches or the scientific realm more important. We should decide about the second, but if we decided about normal Google hits, CSR would dominate the whole ranking.

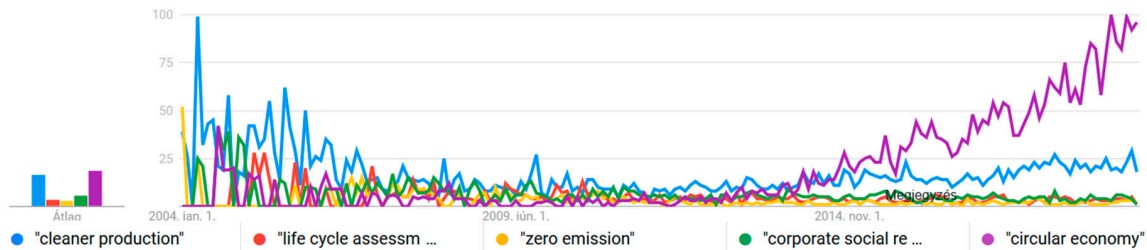

**Figure 13.** Frequency of scientific Google searches for the terms *circular economy, corporate social responsibility, life cycle assessment, cleaner production* and *zero emission*. Data and graph from Google Trends, as of 23 July 2019.

Our analysis can create a basis for a new tool for ranking different business sustainability movements (to be proposed as a future work). If we employ a critical analysis of our results, we can say that Google hit time series is a good first approximation of fashionableness, but does not provide deep scrutiny, compare content of movements, or assess their contribution to sustainability.

## 5. Conclusion: Little Competition, Much Synergy

Google trends is not a 100% precise analytic tool calibrated for scientific analysis, but due to its comprehensive nature, enormous access to data, and ease to use, it is an optimal tool to make quick analyses about an arbitrarily chosen to set of research phrases. Hereby we used it to see the popularity of fifteen business and economic sustainability movements and their change over time.

This approach is fresh but not unprecedented, for instance Denise Reike, Walter J.V. Vermeulen, and Sjors Witjes very recently published an article [56], looking at the Scopus hits of 12 movements similar to circular economy between 1970 and 2016. There is some overlap between the two studies, but apart from *recycling* and *cleaner production*, there are no common terms in the analysis. The reason for that is that we looked at circular economy from a broader perspective of sustainability, the Reika 2018 article is more precise and technology focused. They also used AND analysis, e.g., *circular economy* AND *reverse logistics*. Trends are very similar but focus of the study is also a bit different: we tried to look at life cycle of the business sustainability movements, and whether they can be seen as independent, competing, or symbiotic and mutually reinforcing concepts.

Another line of research does not take such a wide scope but tries to find common and differential points among some of the movements we proposed, for example between the *blue economy* and *circular economy* [57,58], *cleaner production* [59], *environmental accounting* [60] or specific areas of (nonsustainability) management [56]. A very popular line of papers deploys the concept of circular economy for a certain industrial application, like a factory or a domestic industry [58,61].

One of our basic questions were whether the 15 analyzed business sustainability movements are *independent*, *competing*, or *symbiotic and mutually reinforcing*? We have enough evidence to say that they are symbiotic. If we look at the number of publications in Science Direct, we see that all movements are on steep rise in the last 5 years. In other words, our presumed life cycle (on Figure 4) is valid to 80–85% only: till the absolute maximal point on the figure. In reality after that point the trend does not drop, only its acceleration is slower, the curve might level-off or keep on rising, but at a more moderate pace.

The last phase of the trend line does not resemble the falling tail of a Gauss-curve, but a sigmoid curve. In a non-mathematical language: the business sustainability movements live in harmony, they refer to older movements as predecessors, the common field is much bigger, than the differences. I think, this is good news for all, who do not only seek publication credentials, but hope to contribute to make the economic system more ecologically and socially sustainable! We have a strong basis to hope that we do "much ado about SOMEthing".

**Funding:** This research has been supported by the Hungarian National Research, Development and Innovation Office, from the NKFI Fund (grant number K-120044). The APC was funded by the EFOP 3.6.1-16-2016-0007, "Intelligens szakosodási program a Kaposvári Egyetemen" project.

**Conflicts of Interest:** The author declares no conflict of interest. The funders had no role in the design of the study; in the collection, analyses, or interpretation of data; in the writing of the manuscript, or in the decision to publish the results.

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
