# Peer review of "Circular Economy and its Comparison with 14 Other Business Sustainability Movements"

_resources, doi:10.3390/resources8040159_

Round 1

Reviewer 1 Report

The Circular Economy is a current key concept in the shift into a new economic development paradigm. Resources are scarce and finite, the natural systems are increasingly under stress and the mineral deposits are harder to find, of lower grade, more complex (requiring more complex extraction and processing systems) and increasingly difficult to access (physically and, especially, socially). 

I agree with the author's perception that the Circular Economy concept appears in sequence of other related concepts introduced along the last decades in modern economies.

The text proposed for publication falls short, in my view and should be rejected as a journal article:

I don't understand its aim: what is the point of discussing and analyzing the life cycle of the Circular Economy concept? What can be gained from this? The methodology is poorly described, with most of the explanation inserted in a footnote. The work relies heavily on Google Trends and Google search, yet there is no explanation of the science behind it. e.g. what are "normal" Google hits? The author does not describe "circular economy itself, as it is the topic of our article and the whole thematic issue", line 65. How is it possible to leave undescribed the focus of the proposed paper? The paper's research is based only in (Google hits of) English expressions; what happens in other languages and cultures (e.g. Chinese, Russian, French, Spanish or Portuguese speaking)? Are the results similar? Finally, I would avoid mixing Religion and Science. It is never a good idea. As such, the paper's opening paragraph should be omitted or replaced.

It pains me to do so but I can't agree with the publication of this paper given the extreme conceptual flaws detected.

Keep working on this subject; it is of key importance.

Reviewer 2 Report

The author present an interesting work in which it is explained the concept of circular economy. It is adaptable to this journal. I think that the quality is very high, however I must required major revisions. In fact, there are some observations:

title is not adequate. The life cycle concerns other scientific methodologies.

abstact is perfect

please use the typical names of a scientific paper (introduction, materials and methods, results ...)

section 1 can be improved through appropriate literature. CE contributes to the energy perspective https://www.mdpi.com/2076-0760/7/9/148  and material one https://www.sciencedirect.com/science/article/pii/S0959652619315355. In addition, there are high-cited literature review https://www.sciencedirect.com/science/article/pii/S0959652617330718 and https://www.sciencedirect.com/science/article/pii/S0921344917302835 . Finally this journal provides its contribution https://www.mdpi.com/2079-9276/8/1/32 and https://www.mdpi.com/2079-9276/8/2/91

Section 2 is excellent. I think that you can added an initial figure in which all names are collected

Section 3. There is a great risk with the use of life cycle. I think that you analyse the past, present and future of CE approach. The title can reflect this vision.

Section 4 presents results with a good perspective. However, a new tool can be followed: the proposition of a ranking regarding different movements (it is can be proposed as a future work). In addition, it is necessary a critical analysis of results obtained. What are the main aspects obtained by your analysis?

Section 5 is not adequate. Some results must be present in the previous section. Here there is the space only to collect main findings and it is possible to underline limits of your methodology and future directions of research.

I'm sure that following all my suggestions, this paper will be accept during the second step of review.

Author Response

Please find my answers in the attachment!

Round 2

Reviewer 1 Report

Dear colleague,

Thank your comments and willingness to accept or comment most of my suggestions.

I still have doubts on the paper's adequacy in a scientific journal:

What is the point of the article? What knowledge derives from the experiment? Is that knowledge useful (in theory or practice)?

You mention that "It is an interesting question to try to forecast whether it (i.e. Circular economy being one of the most popular topics in the business sustainability arena) will be the same in 2-5 years’ time."

Your paper tries to answer that question. Perhaps I am being too harsh in my judgement: circular economy is, I completely agree, an important concept (an evolution, a generalization, of other related concepts).

the graphs must be improved (size, duplicate legends - Figure 3, scale, variable being represented, etc.).

All the best

A reviewer

Author Response

Please see detailed answers in the attachment!

Reviewer 2 Report

Congratulations.

Author Response

I changed most of what you proposed, added new literature, changed title and headings, etc. I really appreciate your useful comments in the first round and your acceptance now. Thank you very much!

Round 3

Reviewer 1 Report

Dear author,

Yes, there has been an improvement; I have also noted the constructive approach you had in this dialogue.

I have conferred with the academic editor and I agree with his remarks concerning Figures 1,2 and 3 (and I would extend them to Figures 4 and 5) and with his decision to accept his manuscript. 

At this moment, the graphical component of the paper is weakest one; bear this is mind in future papers.

Best regards